# Green Synthesis of Carbon Nanoparticles (CNPs) from Biomass for Biomedical Applications

**DOI:** 10.3390/ijms24021023

**Published:** 2023-01-05

**Authors:** Muhammad Qasim, Andrew N. Clarkson, Simon F. R. Hinkley

**Affiliations:** 1Department of Anatomy, Brain Health Research Centre and Brain Research New Zealand, University of Otago, Dunedin 9054, New Zealand; 2Ferrier Research Institute, Victoria University of Wellington, Wellington 5012, New Zealand

**Keywords:** green synthesis, carbon nanomaterials, biomass, sustainability, biomedical

## Abstract

In this review, we summarize recent work on the “green synthesis” of carbon nanoparticles (CNPs) and their application with a focus on biomedical applications. Recent developments in the green synthesis of carbon nanoparticles, from renewable precursors and their application for environmental, energy-storage and medicinal applications are discussed. CNPs, especially carbon nanotubes (CNTs), carbon quantum dots (CQDs) and graphene, have demonstrated utility as high-density energy storage media, environmental remediation materials and in biomedical applications. Conventional fabrication of CNPs can entail the use of toxic catalysts; therefore, we discuss low-toxicity manufacturing as well as sustainable and environmentally friendly methodology with a focus on utilizing readily available biomass as the precursor for generating CNPs.

## 1. Introduction

Carbon nanoparticles (CNPs) have demonstrated utility in a wide range of biological applications such as imaging, sensing and surface-coating. They enjoy a growing range of applications in the drug delivery of a number of biomolecules such as DNA, antibodies, and proteins. CNPs are considered the ideal candidates for metal-based sensor applications and rapid diagnostic assays due, in the main part, to their high fluorescence value. As such, they can displace gold, colored latex, silica, quantum dots, or phosphor nanoparticles, in relevant applications (Figure 1) [1,2,3,4,5].

CNPs have a range of highly desirable attributes; they exhibit low toxicity, high biocompatibility, can be readily suspended into solution [6] and can be modified post-production with a range of chemical functionality. Carbon based materials already play a critical role in many applications: electro-catalysts, electrodes in storage devices, biofuels, heterogenous catalysis and photo catalysis [7]. Carbonaceous materials form the basis of gas storage (e.g., CO_2_), hydrogen capture, water purification and as additives to rectify soil properties [2]. The field of optical sensing has been extended with the discovery of highly fluorescent CNPs and carbon nanodots (CNDs) [8,9]. CNP and CND probes demonstrate remarkable properties; emission characteristics that are tunable based on particle size, high emission quantum yields, physical and chemical stability, narrow spectral bands and optimization of the surface to effect selective sensing applications [10]. Other than optical sensing, these CNPs have also seen use in applications such as photocatalysis, bioimaging and optoelectronics [11,12,13]. A typical example of optical sensing is the light-emitting properties of semiconductor quantum dots (QDs) used for in vitro and in vivo bioimaging [14,15,16] (Figure 2). QD’s strong optical absorption peaks are very sensitive to the surrounding environment and medium, which enables biomolecule calorimetry to be completed [17,18]. Another significant advantage of CNPs is that they can be produced economically with high purity and are readily fabricated from naturally occurring biomass and generated utilizing green chemistry [19].

Since the pioneering studies by Sumio Iijima in 1991 with the characterization of carbon nanotubes, CNPs have been used in a number of applications. The impact of CNPs has been recognized internationally with researchers receiving the highest of scientific awards [20]. More recently, due to their adoption in the fields of medical, environmental and novel materials, industry has paved the way for the rapid development of various CNPs, such as carbon nanotubes (CNTs), QDs, graphene and nano diamonds (NDs). Based on the lattice arrangement of the main building unit carbon, CNPs are classified as either one-dimensional (1D), two-dimensional (2D) or three-dimensional (3D) structures. For instance, CNTs are 1D, graphene is 2D and NDs have a 3D structure [21,22]. Among these CNPs, CNTs are the hardest materials due to their extended aromatic C-C bond network. The arrangement of these single C-C network layers may be classified further into either single-wall carbon nanotubes (SWCNTs) or multiwall carbon nanotubes (MWCNTs) [23,24]. This C-C bond network gives a honeycomb-like arrangement. SWCNTs and MWCNTs have inter-layer distances of 0.32–0.35 nm and 2 to 50 sheets are found arranged in the tubules. This gives rise to a wide range of wall thicknesses [25]. In the case of graphene, discovered recently in single-layer form as reported by Novoselov [26], a hexagonal arrangement is generated through an extended sp^2^ hybridized network. 

Production of CNPs may be achieved through chemical vapor deposition (CVD), arc discharge vaporization, floating catalytic methods, laser ablation/evaporation and low temperature solid pyrolysis (Figure 2) [27,28,29,30]. However, accepted methods for the synthesis of CNPs have environmental concerns such as the high consumption of raw materials and the use of strong acids. In addition, CNP containing materials produced using these methods exhibits a limited capacity for the loading of metals oxides or post-generation modification of the particle surface [31,32]. Therefore, the focus of CN production has moved towards establishing methods that are less demanding of resources, low-cost and eco-friendly [33]. 

In a green-synthesis approach, biomass such as woods, leaves and low-value biomaterials, such as plant husks, have been used as the precursor for developing carbon materials. In this review we consider that a green-synthesis attribute permits inclusion. For example, a renewable feedstock, process attribute or combination therein. In some cases, green synthesis through biomass as the starting material can also replace the metal catalysts used in CNP fabrication. Based on the reaction conditions and precursor materials, different forms of CNPs such as nanofibers, nanotubes or nanoporous configurations can be generated. For instance, allotropic forms of graphitic carbon nanostructures, with a coil morphology, were fabricated by a precipitation method at 900 °C via the hydrothermal treatment of cellulose [34]. In addition, carbon nanosheets can be fabricated from lignocellulosic biomass derived from coconut coir through the action of hydrothermal carbonization followed by pyrolysis [35]. The effect of clay mineral particles has also been evaluated [36,37]. Similarly, cellulose (husk) has been used to develop carbon hollow nanostructures via a three step acid digestion process, followed by charring and high temperature pyrolysis (CO_2_ laser 2200 °C) [38]. Many other types of food and agricultural waste such as proteins, chitin, lignin, carbohydrates, hemicellulose, and honeycomb have been used in the fabrication of CNPs using a green synthesis approach [19,39,40].

Herein, we describe the methods used for generating CNPs from renewable precursors and highlight some examples of their application in the environmental, energy-storage and medicinal fields.

## 2. Processing of Biomass

Biomass waste can be derived from either plant or animal matter as a result of processing higher-value materials, for instance leaves from trees as by product from processed wood or paper production [41]. Biomass is abundant: trees, agriculture crops, energy crops, fruits, vegetation, wood, aquatic plants and algae, general municipal waste and animal waste [7,42]. In general, such biomass materials are subjected to various processing methods to obtain energy and carbon allotropes. Two broad classifications may be made: biochemical processing (e.g., anaerobic digestion and fermentation) and thermochemical processing (e.g., pyrolysis, combustion and gasification) [43].

The process of pyrolysis can be a highly efficient energy recovery process and has the potential to produce products ranging from char, to gas and oil [44]. Char as a by-product of energy recovery processes can act as a source for various carbon materials such as activated carbon, porous carbon and CNPs such as CNTs, graphene and fullerenes, all of which can be generated through controlled green synthesis processes [45,46,47,48]. These CNPs can be functionalized further and their surface texture and functionality modified by using different surface treatment agents and activation methods [49,50]. Final products find a wide range of applications such as environmental sensors, water purification, hydrogen capture and storage, energy conversion and air pollution control [50,51]. Generally, to obtain CNPs from the by-product of biomass processing three types of treatments are used: physical activation, chemical activation and hydrothermal carbonization (HTC) [20,52].

Physical activation is a two-step process; raw materials are subjected to pyrolysis and carbonization at a temperature below 1000 K, and in second step, subjected to controlled gasification process at high temperatures above 1150 K, in the presence of oxidizing gases (CO_2_, air, steam or a mixture of these) [53]. With steam or CO_2_ as an activation gas, equipment is easy to clean, and the removal of the oxidant is straightforward. Various biomasses such as rice straw, peanut, rice husk, corn hulls, corncob, coconut shells, pecan shell and almond shells have all been used to developed CNPs using this physical activation method [54,55,56,57,58].

With chemical activation, a well-established single step is undertaken where a precursor is mixed with a chemical activation agent (H_3_PO_4_, ZnCl_2_, KOH, K_2_CO_3_, etc.) and when heated to temperatures ranging from 700 K to 1200 K, carbonization and activation occur simultaneously [59,60,61,62]. The chemical activation process results in carbon materials with high porosity and surface area (>2000 m^2^ g^−1^), and larger pore sizes [41]. The chemical activation process has advantages compared to physical activation as it is faster, requires lower conversion temperatures, is higher in carbon yield and provides a more uniformly high-porosity material. Among chemical agents, KOH is favored and since 1978 active carbon produced from KOH treatment processes has generated material with uniform porosity and a high surface area (up to 3000 m^2^ g^−1^) [63]. 

The process of hydrothermal carbonization (HTC) is inspired by natural processes in which biological materials undergo a long, natural chemical coaling process. The application of high pressure and heat converts biomass to peat or coal over thousands to millions of years in a natural phenomenon [40]. HTC is the direct chemical imitation of this natural process but occurring over a much shorter timeframe. This process was first reported by Bergius in 1913 and remodeled by Berl and Schmidt in 1932, which is the well-known methodology for converting cellulose to activated carbon, and is still in common use today [63]. Recently, this method was shown to produce carbon materials from biomass with much milder conditions using temperatures below 500 K, pure water and self-steam pressure [40,49]. This process is considered both physical and chemical processing and is desirable due to the comparatively low temperature, cost effectiveness and overall eco-friendly synthesis. Carbohydrates and their derivatives such as hydroxymethyl furfural, glucose, xylose, furfural, sucrose and starch have all been converted to carbonaceous materials with HTC using a temperature of only 180 °C [64]. HTC can generate porous materials directly from biomass but, as compared to the chemical process, a less porous and lower surface area product results [40,65]. Therefore, such carbon materials are not optimal for the applications of chemical or gas adsorption, or catalysis and energy storage. To improve the porosity of carbon materials from HTC, different templates (e.g., SPA-15) or additives (and therefore more chemical-process aligned methods) such as KOH are used [66,67].

## 3. Carbon Nanotubes (CNTs)

CNTs were discovered in 1990 by Iijima and research into their potential has blossomed in the field of nanotechnology [68]. The unique chemical, optical, physical, thermal conductivity and electrical properties make these particles ideal for sensors, transistors, fuel cells, field emission devices and logic circuits [69,70,71,72,73,74,75,76,77,78]. CNTs and fullerenes are carbon forms that are characterized by their hollow structure. The cylindrical form is known as a nanotube, while spherical fullerenes are known as bucky balls [79,80]. The single-walled carbon nanotubes (SWCNTs) are types of CNTs that are developed by rolling single sheets of graphene into seamless cylinders [81]. Multiwalled graphene sheets are called multiwall carbon nanotubes (MWCNTs) and were discovered by the Russian scientists Radushkeivch and Lukyanovich in 1951 [82], preceding SWCNTs by decades [68,83].

### 3.1. Biomedical Applications of CNTs

CNTs possess unique characteristics which make then an ideal candidate for various applications in biomedicine. CNTs are hydrophobic and therefore are suitable for various therapeutic and diagnostic biological applications [84]. In addition, CNTs have a high drug loading capacity, large surface area, high mechanical strength, and adequate chemical stability, which has made CNTs idea candidates for both therapeutic and diagnostic applications, including being excellent nanocarriers for drug delivery [85]. For instance, CNTs have been shown to interact with receptors present on the cell surface that can result in greater cell loading of drugs and therefore, a reduction in the dose of a drug required to achieve an effect [86].

To date, CNTs have been utilized in a wide variety of drug delivery systems for the treatment of numerous diseases. For instance, substances such as acetylcholine, that does not cross the blood-brain-barrier, can be delivered readily to the brain using CNTs [87]. In addition, because CNTs interact with brain cells they are becoming recognized as being ideal for developing efficient gene and drug delivery systems [88]. In support of this, cell delivery of nucleic acids has become one of the prime functions for the use of CNTs [89]. In addition, groups have also used CNTs (as graphene nanosheets) to deliver angiogenic genes specific for tissue engineering and regenerative medicine to promote revascularization and cardiac repair [90]. Whilst the biological applications for CNTs are vast, much work is needed to validate their use in order to take them into the clinic. 

### 3.2. Synthesis of CNTs

Other than the natural production of CNTs, as discussed earlier, there are many synthetic ways to develop CNTs such as electrolytic methodologies, arc discharge vaporization, laser ablation/evaporation, low temperature solid pyrolysis, chemical vapor deposition (CVD), the floating catalyst methods, and the ion bombardment growth method [27,28,29,30]. All these processes generate CNTs in moderate yield with unique morphology [91]. Some of these methods, such as CVD, laser ablation and arc discharge require an atmosphere of inert gas (such as He, N_2_, or CF_4_) to allow condensation of the nanoparticles to form in the cooler parts of the chamber and to minimize the chances of over-oxidation at high temperature [92].

In order to produce a large quantity of CNTs, CVD of carbon placed in a fluidized reactor at 15–150 kgh is used [93,94]. This multidimensional process controls CNT agglomerate formation, provides delicate catalyst control at the atomic level, and keeps large scale production of CNTs at a macroscopic level [93]. The structure of CNTs can be modulated by the choice of the catalyst that acts at the atomic scale—this is one of most important factors when producing large quantities. Generally, transition metals such as Y, Pt, La, Mo, V, Ni, Co, and Fe are used as a catalyst in the synthesis of CNT [95]. Metal is preloaded into a catalyst carrier (mostly alumina) by a process of impregnation, co-precipitation and also other processes which are common in the petrochemical industry. The calcination reduction method used for metal particle activation during or before CVD treatment gives rise to catalytically active nanoparticles that show dual functionality as a time activator for carbon and as a template for the formation of the CNP product [96].

Metal catalysts have a significant disadvantage in that any entrained catalyst in the production of CNTs pose a significant health threat, particularly if the product destination is for in vivo medical applications [97]. Metal catalyst particle residue in CNTs has resulted in misalignment of plant chromosomes during the metaphase resulting in cell division arrest [98]. In the textile industry, metal residue was not readily removed by water treatment [99]. Nanoparticulate metal catalyst can inhibit florescent attributes of CNTs and so are not suitable for preparing materials for semiconductor based devices as their presence reduced quantum efficiency and detector life [100,101]. During the process of CVD using high temperature, metal particles vaporize and re-condense to the surface of CNTs, intimately coating the CNT surface to block their lattice vibration, and ultimately disrupting the lattice wave propagation [100].

In addition to the aforementioned conventional synthesis methods for CNTs, another important factor is cost: instrumentation, extreme temperature (700–1200 °C) and metal catalysts all contribute to CVD manufacturing economics [102,103]. Therefore, development of non-metallic, cost-effective catalysts, ideally functional and using lower temperatures, will enable better production and commercialization of CNTs to meet the obvious demands in medical, agriculture, textile and electronic fields [104].

### 3.3. Green Synthesis of CNTs

In order to simplify the preparation, and improve the quality and quantity of CNPs, new CVD preparation processes have been developed, which include radio-frequency enhanced CVD, plasma-enhanced (PECVD) and microwave-enhanced methodology [105,106]. In addition, spray pyrolysis has emerged as an alternative process for commercial production of CNTs [107] and is attractive due to the much reduced complexity in processing, the use of low-cost instruments, and it does not require high vacuum or the application of reducing agents. It is also highly amenable to scaled-up commercial production. In combination with a renewable biomass starting material this is a highly attractive methodology. For instance, the addition of wood sawdust in spray pyrolysis resulted in CNPs of good quality and quantity producing both carbon fibers and nanotubes [108]. Hydrogen was produced as a by-product alongside a small amount of toxic gases (NO_2_, CO). Significantly, a low temperature of 750 °C (as compared to CVD to develop CNT/CE that uses a temperature of 1200 °C), resulted in the production of CNPs with a diameter of 50 nm [109]. 

The use of oils in spray pyrolysis has been shown to produce both MWCNT and SWCNTs. For instance, neem oil pyrolysis was used to develop uniformly aligned MWCNTs [110], whereas eucalyptus and turpentine oils produced SWCNTs with a thickness of 0.79–1.71 nm [108,111]. CNTs produced from turpentine oil developed within 60 min at 700 °C, with the size shorter than eucalyptus oil derived materials, while the turpentine oil based CNTs demonstrated a good degree of graphitization and exhibited finely resolved concentric shells. Furthermore, CNTs produced from turpentine oil contained less defects and had a notably higher current density capability and field emission strength [112]. In a similar manner, a castor oil–ferrocene mixture in an ammonia solution generated nitrogen-containing CNTs through spray pyrolysis [113] that resulted in CNTs with a distinctive bamboo shape and a wavy tube-like structure of 50–80 nm thickness. 

### 3.4. Synthesis of CNTs from Natural Precursors

The structure and morphology of CNTs can be regulated by controlling the reaction gas, temperature, catalyst and precursors during their synthesis. Due to the high demand for CNTs worldwide [114], specifically for cancer treatment with graphene NT and medical imaging it is desirable to find eco-friendly precursors for commercial production. Biomass derived natural materials such as camphor powder, palm oil, neem oil, eucalyptus and palm tree provide alternatives to fossil-fuel based precursors such as xylene, acetylene, methane or toluene, etc. [115]. The use of these natural precursors can produce large quantities of CNTs and through the application of appropriate processes may also be generated cost effectively [116].

The elemental analysis of edible oils has shown that they are a viable source for CNTs where the oils solids comprise 73.8% to 77.2% carbon [106]. In a recent report, coconut milk has been used as a precursor for CNP production through a simple and one step process using thermal pyrolysis at 120–150 °C. Particles were produced in only 2–5 min, without the use of any passivating or carbonizing agent [19]. In this process, carbon rich residues are separated from coconut oil by pyrolysis and when dissolved in water, exhibit blue fluorescence under UV light. A similar process was used to develop CNTs from olive oil as carbon precursor and NiCl_2_ as the catalyst at 900 °C [105]. The resultant CNTs were SWCNTs with uniform surface morphology and diameters of ∼27–31 nm [105]. In another study, SWCNTs were produced from vegetable oil as the carbon precursor with CNTs being produced having a diameter of 0.79–1.71 nm [117]. The MWCNTs with an aligned macrostructure of nanotubes were developed when vegetable oil was premixed with ferrocene. 

A crystalline latex that was extracted from *Cinnamomum camphora* (camphor C_10_H_16_O) has been used as the carbon precursor for large-scale production of CNTs [118,119]. Plants of *C. camphora* are very common in the sub-tropical region including Japan, India, China and Indonesia. Thermal decomposition of camphor at 875 °C under argon produced a variety of nanotube that contained a mix of aligned CNTs, as well as multiwall and single wall CNTs [39]. The CNTs that are well aligned vertically are called vertically aligned CNTs (VACNTs) and are 1D carbon objects that are anchored atop of a solid substrate. These VACNTs are geometrically fixed compared to their counterparts, randomly oriented carbon nanotubes (CNTs). With camphor, a minute amount of catalyst was required and no amorphous carbon was produced, meaning no post deposition heat-process was required as is required in thermal decomposition methods [118]. Similar to vegetable oil, chicken fat mixed with ferrocene can be used as the precursor for CNT production [120]. Suriani et al., showed that VACNTs can be generated from chicken fat by using a ferrocene catalyst and deposition of carbon particles onto a silicon wafer substrate [107]. VACNTs demonstrate a highly crystalline structure with a D- Raman peak and G- Raman peak ratio (ID/IG) of 0.63 and in 88.2% purity with a very low amount of amorphous carbon content [107]. Coconut oil has been used as a carbon precursor for developing MWCNTs by using the CVD process [121]. Nitrogen gas was used as the carrier for evaporating the precursor and provided an inert environment. The MWCNTs so produced have a diameter of 80–90 nm (under optimum conditions). Finally, castor oil has also been used for the production of CNTS through spray pyrolysis with argon as the evaporated precursor carrier and with heating to 850 °C [121].

### 3.5. Biomass as the Green Catalyst for CNT Synthesis

Development of green catalyst assisted CNTs’ has obvious advantages. Utilizing the biomass as both the carbon source and catalyst doubly so [103] and can provide metal impurity-free CNTs. Natural green catalysts are available in abundance at negligible cost and have been shown to operate at significantly reduced temperature. There is no requirement of costly equipment such as evaporative sputtering or controlled dip coating when green catalysts are used. Because of the ease of production and non-toxic nature of a green catalyst, there is no requirement for a clean room to optimize CNTs’ growth [104].

CNT production does require a catalyst or a substrate to act as a template [122]. Biomass derivatives can act as that catalyst in CNT fabrication. Iron containing active carbon (AC) is a type of biomass-derivative that can be used as a catalyst in the synthesis of CNTs. Chen et al., demonstrated that wheat straw (AC-W), palm kernel shell (AC-P) and coconut (AC-C) contain iron as an impurity that catalyzes CNT production. If this biomass is pre-reduced with H_2_, the catalyst embedded in AC can be activated [123]. Further, graphitization determines the quality of CNPs if carbon is used. When biomass is used as a carbon precursor, or source of catalyst, or as a catalyst support, then there is a significant saving in time and energy [124]. Botanically derived volatile hydrocarbons are a rich source of carbon and can act as carbon precursors as compared to conventional gaseous precursors [118]. Camphor has been extracted from the latex of the *Cinnamomum camphora* tree and exploited to develop MWCNTs and SWCNTs produced over quartz, zeolite and silicon substrates [118,125,126]. Camphor is less toxic, cost effective and a readily available biomass. Sublimating at around 25 °C it is therefore an ideal candidate for CVD methods in the production of CNPs. When a zeolite catalyst support was impregnated with Fe–Co at atmospheric pressure and temperature of 650 °C, MWCNTs are produce in large quantity and with high purity (88%) from a pure camphor precursor, which made this process suitable for large-scale synthesis [118].

Similarly, walnut (*Juglans regia*), neem (*Azadirachta indica*), garden grass (*Cynodon dactylon*) and rose (*Rosa*) plant extracts have also been used in the production of CNTs, as they act both as source of carbon and as a catalyst due to the presence of active carbon (Figure 3) [104].

The CNTs produced from walnut extract through the CVD process (at 575 °C) are of high quality and good yield when compared to other plant extracts that have been used. Interestingly, when the temperature was raised to 800 °C an increase in another allotrope was observed; carbon nano belts (CNBs) were detected. Their presence was confirmed using scanning electron microscopy, IR and Raman characterization [104]. 

Rice husk (RH) has been used as a carbon precursor as well as the source of catalyst in the production of CNTs under microwave treatment [127,128]. Rice husks are rich in lignin and cellulose, therefore they are an ideal carbon precursor [127]. The introduction of ferrocene enhanced the RH decomposition in the microwave oven induced plasma. It was observed that during this process, ferrocene was converted to non-toxic iron II, III oxide [127]. A major advantage of this green synthesis methodology of CNPs is that either non-toxic metals are produced, or no additional metals are required to be added as catalysts. This method has also been adopted for hybrid synthesis of CNT using polyaniline (PANI), forming CNT/PANI. The nickel electrodes used in the catalytic process are coated by this CNT composite material through surface modulation. This composite material is fabricated by polymerization of aniline in the presence of MWCNT-COOH in a solvent consisting of water and green solvent. The green solvent used is a mixture of ionic liquids (methylimidazolium tetrafluoroborate, [BMIM] BF_4_) and mineral acids such as HCl or HNO_3._ This reaction, which results in CNTs covering the surface of aniline, results in a product that is often used in electronic applications such as chemical sensors [129].

The form and function of CNPs synthesized using a green catalyst methodology is predicated by the composition of the biomass used and processing steps. For example, carbon microspheres (4–6 μm in diameter) were produced when waste cooking oil was used as the starting substrate [130]. Replacing the cooking oil with engine oil generated 4μm spherical CNPs. This difference in size was attributed to engine oils pure hydrocarbon nature, compared to waste cooking oil that is fatty acid derived where stearic palmitic linoleic and oleic acids dominate [131]. Fatty acid decomposition produces more gasses due to the presence of oxygen, which can help in the production of filamentous structures of semi-graphics [132].

Oils are a naturally rich source of fatty acids and in turn carbon, which is used in the production of CNTs. For instance, palm oil from cooking waste has been used for the production of SWCNTs and MWCNTs via the floating-catalyst thermal CVD method [133]. Importantly, the impurities found in these oil wastes do not affect this CNT fabrication process [134].

Over the past 30 years, green chemistry has been a significant focus in the chemical industry, and the use and/or replacement of undesirable solvents one of the central themes. For example, traditional solvents have been substituted by greener solvents as new dispersants for CNT [135], e.g., room-temperature ionic liquids (RTILs). This is because RTILs do not evaporate and therefore are still present at the end of the CNT formation process. With the undesirable catalyst residue dissolved in the solvent the CNT may be recovered through centrifugation or filtration. Recently, deep eutectic solvents (DESs) have been recognized as a new type of low cost RTIL [136]. When RTILs and DESs are combined to form a porogen, they enhance the solubility of DESs, which influence the RTILs’ supra molecular assembly. This may be an advantage for CNT-filled molecularly imprinted powder (MIP) preparations. A CNT-MIP composite was prepared by using RTILs and DESs’ as a binary green porogen system. The RTILs [BMIM] BF_4_ was chosen to stabilize a nanotube dispersion from re-aggregation [137].

## 4. Carbon Quantum Dots (CQDs)

Carbon quantum dots (CQDs) are one of the most important allotropes among CNPs [10]. QDs are usually just a few nanometers in size (1–5 nm) and display a myriad of desirable attributes: high photo stability, high luminescent, broad absorption spectra, low toxicity even compared to other nano-carbon forms, high quantum yield, long fluorescent life, ready surface functionalization with biological molecules, chemical inertness, biocompatibility and high emission tenability [138,139]. With such functionality, CQDs garner interest in applications from energy storage, photocatalytic activity, biosensing, drug delivery, bio-imaging, light emitting diodes, through to their use as fluorescent probes [10,138,139,140,141,142,143,144,145,146,147,148]. In addition, CQDs have a been shown to have wide utility in nanomedicine and biomedical applications [143,148]. CQDs are semiconductor nanocrystals (2–100 nm) and impart unique electrical and optical properties. Due to their single step synthesis, and the above-mentioned qualities as well as water miscible nature, CQDs present as an alternative to traditional fluorescent dyes and inorganic semiconductors. Many environmental, physical and biosensing devices use CQDs and can be used to substitute metal based QDs. 

### 4.1. Biomedical Applications of CQDs

The treatment of many disease conditions requires both spatial and temporal precision for drug delivery. Carbon QDs have the potential here to be superior in that they offer a dual function, that is, they are nanocarriers for specific bioactive compounds in addition they can be used for simultaneous bioimaging. One example is the use of carbon QDs conjugated with doxorubicin which revealed targeted drug release into tumor cells. Further, the fluorescence property of these carbon QDs allowed for image-guided drug delivery [149]. Whilst the update and use of carbon QDs in biological applications has been slow, based on the recently demonstrated utility for targeted drug delivery, there is hope that this will encourage others to use carbon QDs more in biological applications in the future. 

### 4.2. Green Synthesis of CQDs

Biomass has been applied as the carbon precursor, catalyst or catalyst carrier to generate CQDs [19]. Fluorescent CQDs were synthesized by green chemical methods utilizing a wide range of natural resources, for example: pomegranate extract, D-glucose and an aqueous extract from beetroot [150,151]. A cost-effective and environmentally attractive white light emitting material for use in chemical sensing and biomedical applications was prepared through the hydrothermal treatment of pomelo peel [152]. Synthesis of carbon dots has been carried out using naturally available carbohydrates such as glucose, sucrose, citric acid, pomelo peel, and willow bark [151,152,153]. The optical, physical and chemical properties of CNPs are affected by the molecular precursors employed, the specific methodology used, and the pre- and post-treatments carried out.

In another study, a by-product of coconut milk pyrolysis was used for the development of CQDs. Coconut milk is composed of a high proportion of saturated fat (lauric acid comprises ~50% of this oil) which is converted to coconut oil by thermal pyrolysis [154]. A black residue formed during thermal pyrolysis is, in general, discarded, but can be a source of CQDs through further pyrolysis that does not involve any surface passivating (coating to make the materials produced less active) or use of an acidic reagent, with the CQDs being able to be readily dispersed in water. A simple shift in pH and temperature can be used to develop blue wave length emitting CQDs to detect Fe^3+^ ions from coconut milk [154]. Similarly, leaf extracts of neem (*Azadirachta indica*) have been used to develop CQDs by a one-pot hydrothermal treatment [138]. This process was particularly eco-friendly and low cost with the resultant CQDs expressing a high fluorescent quantum yield (up to 27.2%) [155]. These CQDs were used for biosensing, displaying a peroxidase-like-mimetic activity upon oxidation of peroxidase substrate 3,3′,5,5′-tetramethylbenzidine (TMB), when in contact with hydrogen peroxide (H_2_O_2_) (Figure 4). 

This H_2_O_2_ sensitive TMB oxidation property could be exploited for colorimetric detection of H_2_O_2_. In support of this, the amount of ascorbic acid (AA) in solution has been detected by TMB colour change, [155,156] and the assay evaluated using different levels of ascorbic acid from fruits. This low cost and eco-friendly assay based on CQDs set a benchmark for detection of AA in real and complex biological samples. This finds utility with AA a naturally occurring reducing agent, which is highly water soluble and has demonstrated roles in a wide range of human ailments: cancer, Parkinson’s disease, cardiovascular disease, scurvy, mental illness, infertility as well as the common cold [157]. In a similar fashion, Luo et al. reported a CQD system for detecting AA in fruit [156]; the leaves of neem were processed in a one-pot hydrothermal method, devoid of any chemical reagent for passivation, resulting in CQDs uniform in size and of a high florescent value. This study demonstrated a cost effective and eco-friendly conversion of neem leaf to CQDs. 

Similarly, sweet potato peels, cinnamon, red chili, black paper and turmeric were subjected to hydrothermal treatment to produce aqueous fluorescent CQDs with sizes ranging from 3.14 to 4.32 nm [158]. The dose-dependent toxicity (0.1 mg/mL–0.5 mg/mL) was evaluated and the toxicity recorded was dependent on the starting material. Citric acid derived CQDs showed less toxicity as compared to red chili derived CQDs. This difference was attributed to the surface functionality. A similar fluorescent CQD was developed from *Tamarindus indica* leaves by hydrothermal treatment, displaying a quantum yield of 46.6% and were used for Hg^2+^ sensing in the range from 0.01 to 0.1 mM, applicable for biomedical applications [159]. In addition to these studies, edible *Eleocharis dulcis* plant extract underwent hydrothermal treatment to generate water soluble N/P co-doped CQDs (at 90–150 °C) and were utilized in a Fe^3+^ sensor that found utility in an anticounterfeiting application [160].

Ultrasound treatment has been used in the synthesis of CQDs. Laminar CQDs from commercial graphite were produced in a top-down process using ultrasonication (30 min) where *Opuntia ficus indica* extract was used as the catalyst [161,162]. When the same graphite precursor was stirred at low-temperature (50 °C) and for a brief 30 min ultrasonic treatment, small (5–7 nm) CQDs were produced [145]. Another study utilized lemon peel waste developing water-soluble photoluminescent CQDs (size 1–3 nm, quantum yield 14%) with a hydrothermal process that was used for detecting Cr^6+^ (limit 73 nM) (Figure 5). These CQDs were a TiO_2_ composite and demonstrated the photolytic degradation of methylene blue dye where the catalytic activity was enhanced 2.5 times as compared to TiO_2_ alone. This was attributed to a better charge separation at the interface of the composite. Further, sweet potatoes have also been used as the carbon precursor, resulting in the formation of CQDs with a size of 2.0 ± 0.6 nm and quantum yield of 8.9%. The CQDs formed, when coated with (3-aminopropyl) triethoxysilane, provided an environmentally friendly sensing material for detection of oxytetracycline to concentrations as low as 15.3 ng/mL [163].

## 5. Nano-Diamonds (NDs)

Nano-diamonds (NDs) are a new addition to the nanocarbon family, recognizable by a nanosized tetrahedral arrangement of carbon atoms. NDs were discovered accidently when in 1963, modification of carbon through shock compression in a blast chamber generated this allotrope [164]. The detonation process is used for industrial scale production of NDs, where an inert gas or water (ice) filled enclosed chamber is used, which undergoes an explosive pressure change. Based on the gas or liquid coolant it is either a dry or wet synthesis process. This pressure blast process results in a mixture of soot, which consist of 4–5 nm sized diamond particles and other carbo allotropes and impurities [165]. NDs produced as a result of the detonation process exhibit oxygen containing function groups on their surface such as -OH and -COOH. Such chemistry can be used for further functionalization to enhance biological, physical and chemical properties in order to extend their industrial applications [166,167].

The NDs core is made of a sp3 hybridized lattice arrangement with a disordered carbon grouping including sp2 hybridization at the surface [168]. Their characteristic Raman spectrum [169] and low toxicity made them a useful agent for medical diagnostic applications. The diamond core contains lattice defects, which generates fluorescence-emitting colour centers [170]. These colour centers can vary to an extent that the emission covers almost the entire visible spectrum. The centers can be excited with almost any excitation wavelength; the fluorescence emitted is stable and the characteristic of photobleaching is limited. Moreover, the defect centers can be enhanced with high-energy beam treatment followed by thermal annealing [171]. This conveys significant advantage as compared with the molecular dyes that are commonly used in biological imaging. Therefore, the defect-originated colour centers of NDs (after careful optimization) provide an ideal bioimaging tool as an alternative to molecular dyes [172].

The term ND reflects a broad range of diamond-based materials ranging from nanoscale single diamond crystals to a bulk cluster of diamonds [167,173]. There are different classifications of NDs such as nanocrystalline diamond (NCD) and ultra nanocrystalline diamond (UNCD), which is specific to the morphology generated during the crystal growth process. NCD consist of facets less than 100 nm in size, while UNCD describe materials with a particle size of less than 10 nm [174]. NDs are found naturally in interstellar dust, protoplanetary nebulae, meteorites, diamond films and in residues of detonation processes. It has been confirmed that NDs are present in primitive chondrites at a concentration of 1500 ppm along with isotopically anomalous noble gases, hydrogen, nitrogen and other elements. This confirms that natural ND were synthesized before the Sun’s formation and outside of our solar system [175]. NDs, due to their unique optical, thermal, mechanical and electrical properties have a wide range of applications in physical, mechanical and biological systems. In particular, their optical transparency in the form of a diamond thin film, and a high energy band gap in their activation state, make them an ideal candidate for semiconductor applications [176]. In recent studies, various aspects of NDs have been highlighted, such as their role in NCD film fabrication from hydrogen rich and hydrogen deficient plasma, field emission properties, modifying the mechanical behavior of NCD films, chemical and bio functionalization of diamond nanowires, as well as their use for the study of non-covalent interactions [177,178,179].

### 5.1. Biomedical Applications of NDs

NDs have high biocompatibility, and are superior to other CNPs such as CNTs, carbon black or fullerenes, hence they have received significant attention for their use in biomedical applications [180]. NDs high size to surface ratio and easy functionalization with biological molecules give them an edge for both in vitro and in vivo biomedical applications such as drug delivery, single cell imaging, biosensing and protein purification. NDs can emit a specific wavelength of photoluminescence (red or green) when a specific functional molecule is embedded in their crystal lattice [181]. Therefore, they are used as enterosorbents or solid phase carriers in conjugated and non-conjugated form for small and middle size biological molecules such as drugs, proteins (lysozyme) and vaccines [182]. ND use as a probe has been reported by Cheng et al. for the detection of growth hormone receptors in single cancer cell [183]. The growth hormone molecules were covalently linked to carboxylated NDs (100 nm), which were recognized by the A549 cells growth hormone receptors. Similarly, NDs produced through a detonation process were conjugated with bovine insulin applying a physical adsorption process in aqueous solution. This material regulated sodium hydroxide absorption based on pH. When the pH was alkaline, NDs conjugated with a vaccine at a 4:1 ratio and showed 31.3 ± 1.6% adsorption, while at neutral pH demonstrated 79.8 ± 4.3% adsorption [184].

The biocompatibility of NDs during cell division and differentiation in a single cell was assessed by Liu et al. who injected clusters of 100 nm size carboxylated NDs into cells, where cell imaging revealed that they remained in the intracellular environment for an extended time without causing cell damage [185]. It has been shown that NDs are nontoxic during cell division and differentiation thus can be used as a labelling method for tracking cell organelles in stem cells as well as cancer cells [186]. NDs used as fluorescent labels in cells demonstrate the high photostability of the colour emitted as compared to single dye molecules in photoluminescence experiments [186]. In another study, uptake of NDs in cancer and non-cancer cells such A549 human lung adenocarcinoma cells, HFL-1 fibroblast-like human fetal lung cells and Beas-2b non-tumorigenic human bronchial epithelial cells was studied and compared. It was found that in these cells, NDs uptake is mediated by a clathrin-dependent endocytosis mechanism, with healthy cells showing higher update compared to cancer cells [187].

The use of NDs for single cell imaging was reported recently by Mi et al. who showed that when NDs are excited by alpha particles they quickly emit a stable and ultrabright emission [188]. Such types of fluorescent NDs enjoy application in biomedical imaging as biomarkers when functionalized with a nitrogen vacancy (NV) colour center [189], for example; femtomolar detection in immunoassays using magnetic modulation [190]. In addition, nanoscale quantum biosensors have also been used as magnetometers to measure magnetic fields and dipole moments [191,192,193]. Based on the data presented to date, NDs appear to be ideal agents for use as quantum biosensors, as they have good biocompatibility, are sustainable, show stable fluorescence and photostability, and have good coherence time of the NV centers [166]. Another unique trait of NDs modulated with NV centers is that they can release a far red fluorescence which is particularly useful as it differentiates from the auto fluorescent components of cells [194].

NDs use in cellular imaging is extensive, and this carbon allotrope sees additional applications as a drug formulation reagent. Aqueous solubility and membrane permeability have been classified as major factors that limit drug absorption [195]. For instance, sparingly soluble drugs were successfully formulated as demonstrated by Chen et al., where a ND cluster dispersed Purvalanol A and 4-hydroxytamoxifen in water [196]. Therefore, the enhanced aqueous solubility imparted by NDs can be used to generate optimized drug-concentrations in aqueous formulations (Figure 6). This reduces the necessity of complex formulation ingredients or production methods. In addition, the high solubility permits a tuned drug delivery where frequent low-doses (of a highly soluble ND associated drug) are more effective than infrequent high doses (of the poorly soluble drug-form) which is more effective and less likely to drive drug-resistance [197]. 

Further, NDs have also been applied to catalyzed reactions, as oxygen reduction reactions are electrocatalysis by nitrogen-enriched carbon hybridized nanodiamonds (N-doped CND) [198].

### 5.2. Towards the Green Synthesis of NDs

There is a growing field of the biomedical applications of NDs [199] but their synthesis in a green-chemistry or green-precursor sense is very limited, notably being absent in reviews [200,201]—even of those focused on green-synthetic processes [202]. Simple and relatively environmentally friendly synthesis of ND’s from coal in solution by laser ablation has been demonstrated, and while not a renewable resource this is a low-energy path to ND generation [203]. The green synthesis of carbon nanodiamonds by green synthesis presents an opportunity for future development. 

## 6. Graphene

The graphene lattice has a honeycomb-like structure due to the sp^2^ hybridized carbon-carbon network [79] and can be synthesized using both top-down and bottom-up methods. Mechanical and chemical approaches for the synthesis of graphene involve silicon carbide (SiC) for exfoliation of graphite [204], titanium carbide (TiC) [205], tantalum carbide (TaC), unzipping of CNTs [206,207,208], metals such as Co, Ni, Pt, Cu, Ir, Ru [209,210], solvothermal synthesis [211], chemical vapor deposition (CVD) and organic synthesis [212]. Further, graphene oxide (GO) reduction has also been used in the production of graphene [213]. For this reduction process, pre-reduced GO is preferred over non-reduced GO due to its high conductivity. In 1962, Boehm and colleagues were the first to reported low cost and large scale synthesis of graphene through the reduction in GO using hydrazine [212]. The production of graphene through chemical reduction in GO is well established; however, it remains hard to produce large panels of graphene through this process [213,214]. A major limitation is the incorporation of impurities, mainly not fully reduced oxide species, which results in poor electronic properties. Despite extensive advancements in materials sciences, this remains a big challenge to develop a cost-effective process for the high-quality production of graphene. 

Green chemistry is paving a way to solve this challenge to some extent. Ruan and colleagues have reported the use of low cost carbon resources such as food, agriculture waste materials and insects, which has resulted in the production of high quality single layer graphene sheets directly from a copper foil surface when in the presence of argon, without using any purification steps [215]. Similarly, sugar (glucose) in the presence of ammonia solution as the reducing agent, reduced glucose oxide into graphene nano sheet [39]. Similarly, dextran as used by Kim et al. reduced GO for the production of graphene. Due to the biocompatible nature of dextran this process was highly eco-friendly [216]. Triethylamine or *Ginkgo biloba* extract are also reported as reducing and stabilizing agents for green chemical synthesis of graphene. Similarly, the amino acid glycine can also be used as a reducing agent to generate graphene oxide [217,218,219].

Molecular probes or biological molecules interact with GO dynamically allowing the investigation of special biological functions or responses as detected by changes to Raman scattering patterns and unique fluorescence observations [220]. Graphene nanomaterials were effective in the biosensing of genetic materials; e.g., single and double stranded DNA and RNA [221]. Their highly selective biosensing ability is attributed to a strong ionic interaction with positively charged nucleobase to negatively charged carboxyl groups on the GO surface, high electrochemical nature and fast π–π stacking between the nucleobases and honeycomb carbon structure. Using these properties the group of Rahighi fabricated reduced graphene nanowire (RGNW) as a biochip for detection of the four bases of DNA with the potential to detect at the remarkably low sub-femtomol level [222]. Similarly Zhang et al. functionalized GO with -COOH and with polyaniline (PANI) to fabricate two different type of GO such as GO-COOH and GO-PANI to detect DNA, also at very low concentrations (10^−6^ to10^−14^ mol/L) [223]. Graphene nanoparticles have been adopted widely to the field of drug delivery and sustained release applications. Pei et al. delivered the well-studied oncology therapeutic doxorubicin to PEGylated nano graphene oxide (pGO) and showed its sustained delivery of the cytotoxin with a twofold increased anticancer effect [224].

## 7. Conclusions

High carbon content biomass can be used to synthesized CNPs through simple, green, low cost, and time efficient methods, often with minimal additional chemical or solvent components. Fruit waste is an excellent resource, readily available and requires no additional preconditioning. These biomasses may be manipulated further to produced varieties of CNPs according to their applications. CNPs such as CNTs (SWCNTs, MWCNTs), NDs and QDs have been used for delivery biomolecules (drugs, vaccine, mRNA, DNA, etc.), biosensing for diagnosis of cancer and other diseases, as antimicrobial agents (e.g., antibacterial, antifungal and antiviral) and as coating materials. They are also used as composite materials for 3D bioprinting and tissue engineering. These CNPs have wide ranging applications and significance yet there exists significant opportunity to enhance and simplify their production and purification processes to develop more efficient CNPs and CNPs based devices to improve human wellbeing.

## Figures and Tables

**Figure 1 ijms-24-01023-f001:**
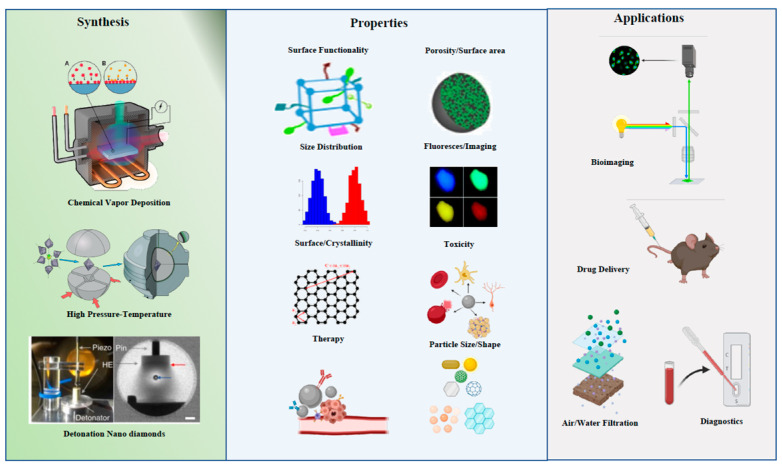
Synthesis, properties and applications of carbon nanoparticles.

**Figure 2 ijms-24-01023-f002:**
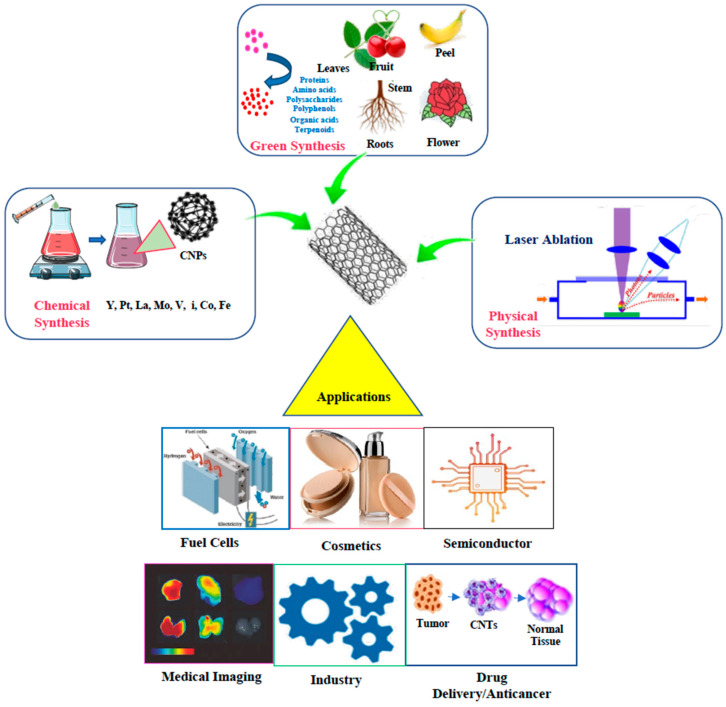
Types of syntheses of CNPs and example applications.

**Figure 3 ijms-24-01023-f003:**
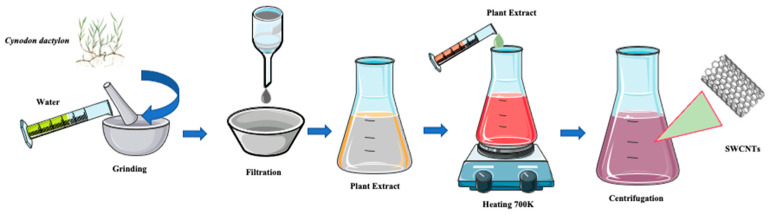
Green synthesis of SWCNTs from garden grass (*Cynodon dactylon*) by the CVD process.

**Figure 4 ijms-24-01023-f004:**
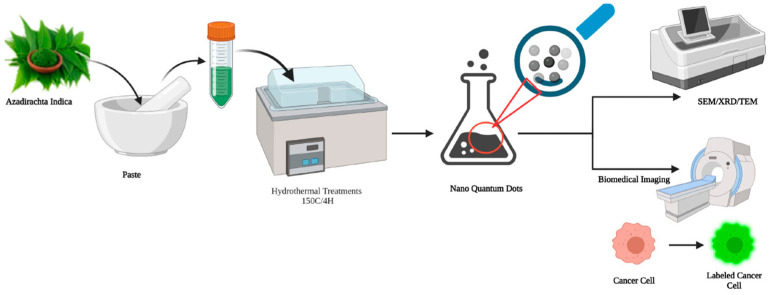
One-pot hydrothermal based green synthesis of nano-CQDs.

**Figure 5 ijms-24-01023-f005:**
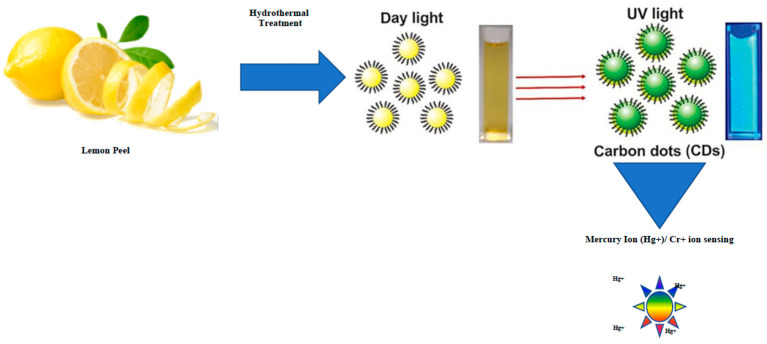
Schematic for the generation of water-soluble CQDs by hydrothermal treatment of lemon peel waste and their application in Hg^+^/Cr^+^ sensing.

**Figure 6 ijms-24-01023-f006:**
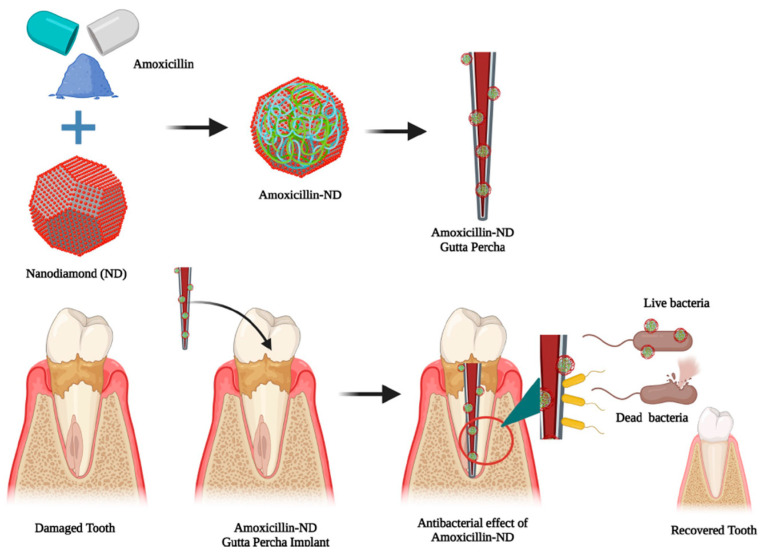
Nanodiamonds (NDs) used in the biological application of amoxicillin-ND aggregates imbedded in gutta-percha as a slow-release antibacterial agent.

## Data Availability

Not applicable.

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
