# Peer review of "Green Synthesis of Carbon Nanoparticles (CNPs) from Biomass for Biomedical Applications"

_ijms, 2023, doi:10.3390/ijms24021023_

Round 1

Reviewer 1 Report (Previous Reviewer 1)

This mauscript can be accepted at its present form.

Author Response

Reviewer 2 Report (Previous Reviewer 2)

The revised version of the review ''Green Synthesis of Carbon Nanoparticles (CNPs) from biomass for biomedical applications'' is looks good, however it still lack the study of biomedical application of carbon nanotubes (CNTs), and carbon quantum dots (CQDs). The author provides only the biomedical applications of Nano-diamonds (NDs).

The author used term graphene in the abstract, but didn’t describe their synthesis methods or applications.

Round 2

Reviewer 2 Report (Previous Reviewer 2)

 The review ''Green Synthesis of Carbon Nanoparticles (CNPs) from biomass for biomedical applications'' is written well and contains a comprehensive knowledge about the green synthesis of carbon nanoparticles. I will recommend this review after minor corrections/revision.

1- Kindly provide notes on Graphene Biomedical application in section-6.

This manuscript is a resubmission of an earlier submission. The following is a list of the peer review reports and author responses from that submission.

Round 1

Reviewer 1 Report

This review manuscript focused on the developments in the green synthesis of carbon nanomaterials, which is interesting topic in the carbon field. However, there are some serious problems should be elucidated before its acceptance.

1. The core term Carbon Nanoparticles (CNPs) should be well defined. In the title, the authors define it as CNPs, but in the abstract, it appeared as Carbon nanomaterials (CNPs). I don’t think it’s reasonable that the Carbon nanomaterials was abbrevatied as CNPs. Furthermore, some of carbon nanomaterials,such as CNTs and Graphene, were hard to be named as CNPs. Maybe it sounds better if they were called as CNMs.

2. The authors should explain or define clearly the “green synthesis”. In my opinion, the green synthesis at least involves the green precursor (raw materials), green preparation technology or process.  

3. The subsection subheading is confused, illogical. For example,

1) in the section 3. Carbon Nanotubes (CNTs), the subheading is as follows:

2.1. Synthesis of CNTs

2.2. Green Synthesis of CNTs

2.3. Synthesis of CNTs from natural precursors

2.4 Biomass as green catalyst for CNT Synthesis

What’s the relationship between these subheadings? I think it’s messy, unclear.

2) 4. Carbon quantum dots (CQDs)

  4.1. Green synthesis of CQDs

There is only 4.1. where is 4.2 and/or 4.3, etc.?

3) 5. Nano-Diamonds (NDs)

5.1. Applications of NDs

There is also only 5.1 without 5.2 or 5.3.

Even worse, there is no synthesis, only application. On the other hand, the title of this manuscript is Green synthesis.

4. There are many other problems, the authors should find and correct by themselves.

Reviewer 2 Report

The review ''Green Synthesis of Carbon Nanoparticles (CNPs) from biomass for biomedical applications'' is written well and contains a comprehensive knowledge about the green synthesis of carbon nanoparticles. I will recommend this review after major corrections/revision.

1- This review is based on synthesis and its application. The synthesis part is discussed in better way but their application side is very week. Author must provide the detail biomedical application with relative literature study.

2- There are few minor grammatical, spell mistakes to be corrected.

3- Author should provide reference and permission for all the figures/images.

4- Author must provide the relative literature study and plant based synthesis about the Nano-diamonds (NDs).

5- How the plant based CNPs are better than chemically synthesized CNPs? How the biomedical application of plant based CNPs will differ from chemically synthesized CNPs?

6- How we can synthesized functional group specific (like N-doped CNPs) via plant based CNPs.?